# Preparation, Identification and Application of β-Lactoglobulin Hydrolysates with Oral Immune Tolerance

**DOI:** 10.3390/foods12020307

**Published:** 2023-01-09

**Authors:** Linghan Tian, Qianqian Zhang, Yanjun Cong, Wenjie Yan

**Affiliations:** 1Beijing Higher Institution Engineering Research Center of Food Additives and Ingredients, College of Food and Health, Beijing Technology and Business University, Beijing 100048, China; 2College of Biochemical Engineering, Beijing Union University, Beijing 100023, China

**Keywords:** cow milk allergy, β-lactoglobulin, hydrolysate, oral immune tolerance, T cell epitope

## Abstract

To reveal, for the first time, the mechanism of T cell epitope release from β-lactoglobulin that induces oral immune tolerance, a strategy for the prediction, preparation, identification and application of β-lactoglobulin hydrolysate with oral immune tolerance was established using the bioinformatics method, hydrolysis, mass spectrometry, T cell proliferation assays and animal experiments. Some T cell epitope peptides of β-lactoglobulin were identified for the first time. The hydrolysates of trypsin, protamex and papain showed oral tolerance, among which the hydrolysates of protamex and papain have been reported for the first time. Although the neutral protease hydrolysate contained T cell epitopes, it still had allergenicity. The mechanism behind oral immune tolerance induction by T cell epitopes needs to be further revealed. In addition, the trypsin hydrolysate with abundant T cell epitopes was added to whey protein to prepare the product for oral immune tolerance. Overall, this study provides insights into the development of new anti-allergic milk-based products and their application in the clinical treatment of milk allergies.

## 1. Introduction

The emerging health hazards of food allergies have increased the demand for exploring effective methods to prevent and treat food allergies. Currently, prevention methods mainly include avoidance therapy, food processing and specific immunotherapy [1]. Avoidance therapy is the method of preventing the intake of food allergens. Food processing is the method of utilizing processing methods to change the structure of food allergens or deactivate their activity. The allergenicity cannot be eliminated completely. Based on the administration route, specific immunotherapy can be divided into subcutaneous immunotherapy, oral immunotherapy (OIT) and sublingual immunotherapy. In the area of food allergies, most studies focus on oral immunotherapy, a therapeutic mechanism inducing the relevant lymphoid tissues of the body, leading to the proliferation of regulatory T cells (Tregs) and inhibition of the immune response, thereby inducing immune tolerance to allergens [2,3]. Previous research on the OIT of milk showed a 37~57% desensitization effect [4,5,6]. Oral tolerance is a state of systemic unresponsiveness, which is the default response to food antigens. Oral immunotherapy with whole food allergen protein has achieved some progress in controlling major food allergies, such as peanuts and milk [7,8]. However, this therapy might cause allergy sufferers to develop severe allergic reactions rather than tolerance. Since the whole allergens contains both T cell and B cell epitopes, the latter is considered as a peptide which can induce an allergic reaction [9]. To avoid a potential allergic reaction, specific allergens containing T cell epitopes or their derived peptide sequences could be applied, which may induce immune tolerance [10] when they are accepted by the antigen-presenting cells in the intestinal tract.

β-lactoglobulin (β-LG) is a major allergen in milk, accounting for about 50% of whey protein and 10% of total milk protein; however, about 82% of the general population suffering from a cow milk allergy is allergic to β-lactoglobulin [11]. The vast majority of cow milk allergies are IgE-mediated type I hypersensitivity reactions, and symptoms appear within 1 h (rapid onset) or 24–72 h (delayed onset) after ingestion of milk. Patients with such anaphylaxis usually have a series of clinical symptoms, most patients have at least two organ systems with allergic symptoms, about 50% to 70% of patients have skin allergy symptoms, 50% to 60% of patients have gastrointestinal symptoms and 20% to 30% of patients have respiratory symptoms [12]. There are many studies on the B cell epitopes of β-lactoglobulin, but the mechanism of β-lactoglobulin releasing T cell epitopes by protease to induce oral immune tolerance in allergic patients need to be explored. Therefore, preparation and identification of β-lactoglobulin hydrolysates with oral immune tolerance of T cells is of great significance to ensure the safety of the milk allergy population.

In the present study, firstly, the T cell epitopes of β-lactoglobulin were predicted by the bioinformatics method, and then, to screen the protease, the hydrolysis conditions were optimized to allow β-lactoglobulin to release more T cell epitope peptides. Secondly, amino acid sequences of β-lactoglobulin hydrolysates were identified by mass spectrometry, and then, some peptides were synthesized using the solid phase synthesis method and identified by T cell proliferation assay. Finally, animal experiments were carried out to identify the immune tolerance of β-lactoglobulin hydrolysates.

## 2. Materials and Methods

### 2.1. Prediction of T Cell Epitopes of β-LG

The amino acid sequences of β-LG were obtained from GenBank and T cell epitopes were predicted by the IEDB online website tool. The possible T cell epitopes of β-LG were predicted based on the HLA-DRB1 and HLA-DQB1 alleles, i.e., DRB1*08:01, DRB1*08:02, DRB1*08:03, DRB1*08:04, DRB1*08:09, DQB1*04:01 and DQB1*04:02. Similarly, the predicted result was obtained by inputting the amino acid sequences of β-LG in the NetMHC(II)/pan4.0 website. The higher the score, the more likely it is a T cell epitope of β-LG.

### 2.2. Preparation of β-LG Hydrolysates

B-LG (Sigma Company, MI, USA. Purity ≥90%) was prepared as a 3% (*w*/*v*) aqueous solution. The optimum temperature and pH of six kinds of protease (Novozyme Company, Copenhagen, Denmark) are as follows: the optimum pH of neutrase, papain, protamex and protease M is 7, while the optimum pH of trypsin and alcalase is 8. The optimum temperature for papain, protamex and alcalase is 60 °C, while the optimum temperature for neutrase, trypsin and protease M is 50 °C. The addition amount of protease was 3000 U/g proteins. The hydrolysis time was three hours. During protein hydrolysis, the temperature and pH of the reaction system were maintained constant. After hydrolysis, the enzyme was inactivated immediately in a water bath at 85 °C for 10 min.

### 2.3. Determination of the Degree of Hydrolysis (DH) and Peptide Content

The DH of the hydrolysate was determined by the OPA method [13]. Simply, a standard curve was plotted using L-leucine. An amount of 50 mmol/L of methanol solution (10 mL), 50 mmol/L of N-acetylcysteine distilled water solution (10 mL), 20% (*w*/*v*) SDS (5 mL) and 0.1 mol/L of borate buffer (75 mL, pH 9.5) were mixed to prepare the OPA reagent. An amount of 3.2 mL of the freshly prepared OPA reagent was taken and put in a test tube and then added with 400 µL of hydrolysate, i.e., the sample or L-leucine standard, and mixed them uniformly. After standing the mixture at room temperature for 10 min, the absorbance of the sample or standard was measured at 340 nm.

The peptide content of each hydrolysate was determined using the biuret method [14] and the reaction conditions were modified. Briefly, 2.5 mL of the hydrolysate was mixed with a trichloroacetic acid solution (10%, *w*/*v*) at a ratio of 1:1 (*v*/*v*). After standing for 20 min, the suspension was centrifuged at 4000 r/min for 10 min, and the obtained supernatant was used to identify the peptide content. An amount of 6 mL diluted sample was mixed with 4 mL biuret reagent. The mixture was shaken well and placed in a water bath for 30 min at 25 °C. The absorbance was read at 540 nm on a spectrophotometer (TU-1810, Persee General Co., Ltd., Beijing, China). Gly-Gly-Tyr-Arg tetrapeptide was set as the standard curve.

### 2.4. Analysis of Amino Acid Sequences and T Cell Epitopes of Hydrolysate Peptide by Mass Spectrometry and T Cell Proliferation Assay

A Q Exactive HF-X mass spectrometer (Thermo Fisher Scientific instrument Co., Ltd., Shanghai, China) and Nanospray Flex (ESI) ion source were used. The ion spray voltage was set at 2.3 kV and the temperature of the ion transfer tube was set at 320 °C. The data-dependent acquisition mode was chosen for mass spectrometry. The full scanning range of mass spectrometry was *m*/*z* = 300~1700 and the primary mass spectrometry resolution was set at 70,000. The maximum capacity of the C-trap was 1 × 10^6^ and the maximum injection time of the C-trap was 50 ms. The parent ion with ion intensity of TOP20 in full scanning was broken by high energy collision cleavage (HCD) and detected by secondary mass spectrometry. The maximum capacity of the C-trap was 2 × 10^5^ and the maximum injection time of the C-trap was 50 ms. The fragmentation collision energy of the peptide was set at 28%, the threshold intensity was set to 2.0 × 10^4^ and the dynamic exclusion range was 30 ms. The protamex cutting site was serine; the alcalase cutting site was serine, glycine, tyrosine, phenylalanine and tryptophan; the protease M cutting site was aspartic acid and glutamine; the neutrase cutting site was tryptophan, phenylalanine, valine and leucine; the papain cutting site was leucine, glycine, lysine and arginine; and the trypsin cutting site was arginine and lysine.

Subsequently, sequences of hydrolysate peptide determined by mass spectrometry were compared with the predicted T cell epitopes. Some peptides were synthesized by the solid phase synthesis method [15]. Then, the T cell proliferation method was used to determine the T cell epitope of the synthetic peptide based on the modified methods of Hou et al. [16] and Joost et al. [17]. In brief, the peripheral blood T lymphocytes of six BALB/C mice sensitized with β-lactoglobulin were separated and cultured in vitro. The T cells were incubated with the synthetic peptides for 48 h. The number of active proliferating cells was detected by an MTT assay kit and the stimulation index (SI) was calculated, which is the ratio of the number of T cell proliferation caused by the peptide to that of T cell proliferation without the peptide. An SI ≥ 2 is a positive reaction, which means the peptide is a T cell epitope peptide.

### 2.5. Oral Tolerance of β-LG Hydrolysate Identified by Animal Experiment

#### 2.5.1. Animal Experiment Scheme

A total of 60 female Balb/c mice (3~4 weeks old, Beijing Weitonglihua Company, Beijing, China) were adaptively fed for 3~4 days in the SPF standard animal room. During the experiment, the mice had free access to food and water (without allergens) and the ambient temperature was controlled at 23 ± 3 °C with a humidity of 40~70% and 12 h during the day and night, respectively. The mice were randomly split into 6 groups, with 10 mice in each group. The negative control (NC) group and the positive control (PC) group were given intragastric administration once a day on day 0, 7, 14, 21 and 28. The intragastric dose of the NC group is 0.3 mL saline and 10 µg adjuvant cholera toxins (CT), and for the PC group, it was 0.3 mL saline containing 5 mg/kg β-lactoglobulin and 10 µg CT. On day 35, the mice were stimulated with 5 to 10 times the dose of saline or β-lactoglobulin without CT in both the NC and PC groups. The four preventive groups were named as the trypsin hydrolysate (TH) group, papain hydrolysate (PH) group, protamex hydrolysate (PM) group and neutrase protease hydrolysate (NPH) group. The mice were administered with four kinds of protease hydrolysates by gavage once a day for 7 consecutive days before sensitization (from day 7), with 20 mg/kg/day (dissolved in normal saline) given each time. On day 0, 7, 14, 21, 28 and 35, β-lactoglobulin was administered for the mice in accordance with the sensitization and anaphylactic provocation phase protocol of the positive control group, shown in Appendix A. All mice were cared for in accordance with the Guidelines for the Care and Use of Laboratory Animals published by the U.S. National Institutes of Health (NIH Publication 85-23, 1996), and all experimental studies were conducted under the program approved by the animal ethics committee of Beijing United University (Laboratory Animal Use License No: SYXK(Jing)2017-0038).

During the experiment, the growth state of the mice, including the physical characteristics, diet, mental state, etc., were observed every week. After high-dose stimulation, the state of mice in each group was observed for about 45 min continuously with the sensitization symptoms scored as follows: 0 points: no abnormal symptoms; 1 point: scratches on nose or head; 2 points: edema around the eyes and mouth, erect hair, decreased activity or increased respiratory rate; 3 points: asthma and dyspnea; 4 points: convulsion or rest after big stimulation; 5 points: death. 

After high-dose stimulation, blood was collected from the inner canthus vein in the eyes of the mice in each group, added into the centrifuge tube with or without EDTAK2 and mixed gently. After storage overnight at 4 °C, the blood was centrifuged at 5000 r/min for 10 min at 4 °C and the serum or plasma was extracted the next day. Then, the samples were stored at −20 °C to determine the specific antibodies, cytokines, chitinase-3-like protein 1 and histamine. Additionally, the level of spleen cell subsets was determined.

#### 2.5.2. Determination of Specific IgE, IgG_1_ and IgG_2a_

The specific IgE level is a key indicator to characterize the success of the animal allergy model. Specific IgE, IgG_1_ and IgG_2a_ were determined by the ELISA method [18]. In brief, β-LG was taken as the coating antigen and 50 mmol/L carbonate buffer solution (pH 9.6) was utilized to dilute the β-LG solution to 10 µg/mL. The mouse serum was added and diluted with antibody diluent (PBST solution containing 1% bovine serum albumin) at a ratio of 1:1000. The HRP-sheep anti-mouse IgE or HRP-sheep anti-mouse IgG_1_ or HRP-sheep anti-mouse IgG_2a_ was diluted 4000 times with antibody diluent. Color development was carried out at 450 nm within 30 min. The absorbance of the positive serum was significantly different from the negative control group.

#### 2.5.3. Determination of TH1 and Th2 Cytokines

IL-4, IL-5, IL-13, IL-17 and IFN-γ were detected using an ELISA kit (Abcam Company, Cambridge, UK) according to the manufacturer’s instructions.

#### 2.5.4. Determination of Histamine

A histamine ELISA kit (Abcam Company, UK) was adopted for determination of histamine according to the manufacturer’s instructions.

#### 2.5.5. Determination of Chitinase-3-like Protein 1 Content 

A chitinase-3-like protein 1 ELISA kit (Abcam Company, UK) was used for determination of chitinase-3-like protein 1. 

#### 2.5.6. Determination of Cell Subsets

Splenic lymphocytes subsets were measured according to Pap et al. [19]. Splenic lymphocytes were collected after high-dose stimulation, the mice in each group were sacrificed with cervical dislocation, soaked in 75% alcohol for 2 min and transferred to a clean table to open the abdominal cavity. Spleen tissues were taken, cut into pieces and then placed on a 200-mesh screen. The inner core of a 5 mL syringe was used for gentle grinding and 2 mL of RPMI1640 incomplete culture medium was continuously added to the tissue until most of the cells in the tissue were separated. The cells were gathered to centrifuge at 2000 r/min for 5 min and the supernatant was removed. Later, 2 mL of red blood cell lysate was added to the spleen cell precipitate with gentle blowing and mixing. Then, lysis was performed at room temperature for 2 min until the red blood cells were completely broken, followed by centrifugation at 2000 r/min. Afterwards, the supernatant was ruled out. An amount of 2 mL of RPMI1640 deficient medium was added to resuspend the cell precipitate, followed by centrifugation at 2000 r/min for 5 min. After washing once, the supernatant was ruled out. An amount of 1 mL of RPMI1640 deficient medium was added to resuspend and 15 µL of the cell suspension was taken for cell counting.

An exact amount of 50 µL of cell suspension was added to FITC-anti mouse CD4 monoclonal antibody and PE-anti mouse CD25 monoclonal antibody or their isotype control antibody. After being incubated in the dark for 30 min, the antibody was washed once with flow buffer. Then, the breaking agent was added and incubated for 30 min, followed by centrifugation. The APC-anti mouse Foxp3 monoclonal antibody was supplemented by incubate for 30 min. After washing once, 100 µL of flow buffer was added to resuspend the cells. Then, the level of Treg cells was monitored by FACSCalibur flow cytometry (BD Company, Piscataway, NJ, USA). 

An exact amount of 50 µL of cell suspension was taken and added to PE-anti mouse CD69 monoclonal antibody, anti-mouse CD183/CXCR3 monoclonal antibody or its isotype control monoclonal antibody. After being incubated for 30 min, the level of effector Th1 cells was detected by flow cytometry.

An exact amount of 50 µL of cell suspension was taken and added to FITC-anti mouse CD69 monoclonal antibody, PE-anti mouseT1/ST2 monoclonal antibody or their isotype control. The level of effector Th2 cells was detected by flow cytometry.

#### 2.5.7. Preparation and Identification of Whey Powder with Oral Immune Tolerance 

β-LG trypsin hydrolysate was selected and added to 3% (*w*/*v*) whey protein aqueous solution at a volume ratio of 1:2 and a dry powder was prepared by spray drying.

The oral immune tolerance function of the tryptic hydrolysate of β-LG (TH) added to whey protein concentrate (WPC) was identified by animal experiments. After 3–4 days of adaptive feeding, the BALB/C mice were fed with whey protein concentrate with or without β-LG trypsin hydrolysate once a day for 7 consecutive days before sensitization (from day 7) at 50 mg/kg/day (dissolved in normal saline) by gavage each time. On days 0, 7, 14, 21 and 28, each mouse was given 5 mg/kg of β-LG (containing 10 μg CT adjuvant) by gavage once a day. On day 35, after overnight fasting, the mice were given 5~10 times the dose of β-LG by gavage. After high-dose stimulation, the mice were killed by cervical dislocation. Jejunum tissue was taken from 8~10 cm below the stomach, fixed in formalin solution and embedded in paraffin. Routine HE staining was performed to observe the pathological changes in the organs.

The apoptosis of intestinal villi epithelial cells was detected using a public in situ apoptosis cell detection kit. Measurement of TUNEL was performed as follows: each slice was counted with more than 50 cells under 400 visual fields. Three sections were selected from each mouse, and 10 mice were included in each group. The number of apoptotic cells that presented positive were transformed into the apoptosis index (AI). 

AI = the number of positive cells in the visual field/the total number of cells in the visual field × 100%. 

#### 2.5.8. Statistical Analysis of Data

The experimental data were statistically analyzed by Microsoft Excel 2003 and SPSS 13.0, and the data were expressed as mean ± standard deviation (SD). Analysis of variance (ANOVA) was performed on the data for each group, and Duncan’s range test was used to compare the mean values. If *p* < 0.05, the values were considered significantly different.

## 3. Results and Analysis

### 3.1. Prediction of T Cell Epitopes of β-LG 

The T cell epitopes of β-LG obtained by the online website tool of IEDB mainly focused on three amino acid sequences, i.e., AA26~46, AA110~128 and AA154~162 (Appendix A). The possible T cell epitopes of β-LG predicted by the online tool NET MHC(II)/pan4.0 were concentrated on four amino acid sequences, i.e., AA13~29, AA32~46, AA55~69 and AA83~108 (Appendix A). The combined prediction results of both tools suggested that the possible T cell epitope region of milk β-LG was on these six amino acid sequences, i.e., AA13~29, AA26~46, AA55~69, AA83~108, AA110~128 and AA154~162.

### 3.2. The DH and Peptide Content of β-LG Hydrolysate

The DH of each hydrolysate was measured by the OPA method (Appendix A). Protease M and alcalase exhibited higher DH values of 45.66% and 44.81%, respectively, while the DH of neutrase and protamex was lower, with values of 10.19% and 13.44%, respectively. DH values of papain hydrolysate and trypsin hydrolysates were 34.95% and 20.61%, respectively. 

The peptide contents of protease M, alcalase, neutrase, protamex, papain and trypsin hydrolysates were 485.12 ± 0.45 mg/g, 494.89 ± 1.24 mg/g, 515.78 ± 0.45 mg/g, 516.02 ± 0.78 mg/g, 516.23 ± 1.56 mg/g and 517.04 ± 0.85 mg/g, respectively, determined using the biuret method, and the contents of peptides in different hydrolysates were not significantly different (*p* > 0.05).

### 3.3. Analysis of Amino Acid Sequences and T Cell Epitopes of Hydrolysate Peptide by Mass Spectrometry and T Cell Proliferation Assay

As shown in Table 1, among the six proteases, ten peptides of trypsin hydrolysate partially or completely overlapped with the predicted T cell epitopes, while four peptides of protamex hydrolysate partially or completely overlapped with the predicted T cell epitopes. As for the neutrase hydrolysate and papain hydrolysate, the numbers of overlapped peptides were four and one, respectively. Polypeptide was not detected in alcalase hydrolysate or protease M hydrolysate at the hydrolysis sites. The peptides which were partially or completely duplicated with the predicted T cell epitopes were preliminarily identified by T cell proliferation assays. Except for peptides with KALPMH (AA157~162), all other polypeptides may be considered as T cell epitopes according to the stimulation index (SI). The peptides in the hydrolysate were all analyzed by mass spectrometry, and only the primary mass spectrum of VLVLDTDYKK polypeptide is shown in Figure 1A due to limited space. Therefore, the hydrolysates of trypsin, protamex, neutrase and papain will be verified through animal experiments.

### 3.4. Clinical Symptoms of Mice in Each Group

The clinical symptoms of mice in each group are given in Table 2. None of the ten mice in the NC group presented allergic symptoms. However, six mice in the PC group presented asthma and dyspnea symptoms, three mice remained motionless, exhibiting symptoms of trembling and muscle contraction and one mouse died. In the TH group, nine mice presented no symptoms and one mouse scratched its nose or head, showing a mild allergy. As for the PM group, eight mice presented no symptoms and two mice scratched their noses or heads, showing a mild allergy. In the PH group, eight mice presented no symptoms and two mice scratched their noses or heads, showing a mild allergy. In the NPH group, two mice presented the symptoms of edema around their eyes or mouths, with erect hair, decreased activity or increased respiratory rate; six mice presented asthma, dyspnea and other phenomena; and two mice remained motionless or showed the symptoms of trembling and muscle contraction. There were no significant differences in symptom scores between the NPH group and PC group. The symptom scores of the TH group, PM group and PH group were significantly lower than that of PC group (*p* < 0.01).

### 3.5. Determination of Optical Density (OD) of Specific Antibodies IgE, IgG_1_ and IgG_2a_ of β-LG

As depicted in Figure 1B–D, the OD values for specific antibodies IgE, IgG_1_ and IgG_2a_ in the serum of PC and NPH groups were significantly higher than the NC group (*p* < 0.05), but the TH, PM and PH groups showed no significant difference from the NC group (*p* > 0.05), indicating that β-LG hydrolysates of trypsin, protamex and papain significantly prevented the allergic reaction of mice, and the oral tolerance of mice was produced after 7 days of continuous intragastric administration. However, the neutrase hydrolysate did not significantly prevent the sensitized reaction in mice.

### 3.6. Determination of Th1-Related Cytokines

As depicted in Figure 1E,F, the levels of IFN-γ and IL-17 in the TH, PM and PH groups were significantly higher than in the PC group (*p* < 0.05), indicating that the occurrence of allergy in mice was actively inhibited and immune tolerance was achieved in the TH, PM and PH groups. The NPH group showed no significant difference from the NC and PC groups (*p* > 0.05), suggesting that the neutrase hydrolysate did not induce immune tolerance in the mice.

### 3.7. Determination of TH2 Cytokines

As depicted in Figure 2A–C, the levels of IL-4, IL-5 and IL-13 in the TH, PM and PH hydrolysates groups were significantly lower than in the PC group (*p* < 0.05), while those in the NPH group showed no significant difference compared to the PC group (*p* > 0.05). The results indicated that the enzymatic hydrolysates of the TH, PM and PH groups did not induce any allergic reactions in the mice, but the enzymatic hydrolysates in the NPH group did induce an allergic reaction in the mice.

### 3.8. Determination of Histamine Release

Histamine exists in the mast cells or basophils of blood. When an allergic reaction occurs, large quantities of histamine are released from these cells.

As depicted in Figure 2D, the histamine levels of mice in the TH, PM and PH groups exhibited no significant difference (*p* > 0.05) compared to the NC group, while they exhibited significant differences (*p* < 0.05) compared to the PC group, suggesting that the hydrolysates of β-LG in the TH, PM and PH groups could inhibit the release of histamine in the allergic mice, thereby successfully inducing immune tolerance of the mice. However, the NPH group showed no significant difference compared to the PC group (*p* > 0.05), indicating that the hydrolysates of β-LG in the NPH group failed to induce immune tolerance, causing an allergic reaction in mice.

### 3.9. Determination of Chitinase-3-like Protein 1 

In allergic reactions, chitinase can regulate the expression of TGF-β and the number of Foxp3^+^ Tregs to alleviate allergic inflammation.

As depicted in Figure 2E, the levels of chitinase-3-like protein 1 released by mice in the TH, PM and PH groups presented no significant difference compared to the NC group (*p* > 0.05), but the levels were significantly lower than those in the PC group (*p* < 0.05). This indicated that the β-LG hydrolysates of the TH, PM and PH groups could inhibit the release of chitinase-3-like protein 1 in allergic mice, thereby successfully inducing immune tolerance in mice. However, there was no significant difference between the NPH and PC groups (*p* > 0.05), suggesting that the β-LG hydrolysates in the NPH group failed to induce immune tolerance.

### 3.10. Determination Results of Spleen Cell Subsets

#### 3.10.1. Differentiation of TH1 Cell Subsets

As depicted in Figure 3A,B, compared with the NC group, no significant difference was observed in the PC and NPH groups, while the levels of the Th1 cells in the TH, PM and PH groups were significantly higher than those in the NC group, indicating that the number of TH1 cells in these three groups increased significantly.

#### 3.10.2. Differentiation of TH2 Cell Subsets

An immediate food allergy is also called a Th2-type allergic reaction. As illustrated in Figure 3C,D, the mice in the PC group presented the most intensive Th2-type immune reactions. There were no significant differences between the PM hydrolysate group and the NC group. Levels of Th2 cells of the TH, PH and NPH groups were significantly higher than the NC group and significantly lower than the PC group.

#### 3.10.3. Differentiation of Treg Cell Subsets

As depicted in Figure 3E,F, compared with the mice in the PC group, the levels of Tregs in the TH, PM and PH groups were significantly increased, indicating that the relative number of CD4+ CD25+ Foxp3+ Tregs could be significantly increased by oral administration of the β-LG hydrolysates of these three proteases before sensitization. The number of T cells in the PC and NPH groups was not significantly increased.

### 3.11. Identification of Trypsin Hydrolysate with Oral Immune Tolerance 

#### 3.11.1. Observation Results of Intestinal Histopathology

To evaluate the oral immune tolerance of β-LG trypsin hydrolysate (TH), it was added to whey protein concentrate for animal experiments. The HE staining results of the small intestines of the mice are depicted in Figure 4A–D. The mice in the PC group and whey protein concentrate (WPC) group exhibited edema and inflammatory reactions, while the TH group did not show edema or inflammatory reactions. 

#### 3.11.2. Apoptosis Results of Intestinal Villi Epithelial Cells

Apoptosis indexes of the small intestinal epithelial cells in the PC, NC, TH and WPC groups were 41.23%, 2.45%, 7.86% and 19.12%, respectively. As depicted in Figure 4E–H, no significant difference was found between the TH group and the NC group (*p* > 0.05), but there was a significant difference between the TH group and the PC group (*p* < 0.05). A significant difference was demonstrated between the WPC group and the NC group (*p* < 0.05), indicating that β-lactoglobulin trypsin hydrolysate could inhibit allergic reactions of mice.

## 4. Discussion

Many peptides with potential biological activities have been discovered in animal protein, most of which are isolated from milk products or released from hydrolyzed milk proteins [20,21]. Enzymatic hydrolysis is usually conducted under mild conditions (pH, temperature, substrate concentration and enzyme activity), and is the most common method to produce oral tolerance peptides [22]. Enzymes without organic solvents and toxic chemical residues have higher specificities, making enzymatic hydrolysis the optimum method for producing active peptides in the food and pharmaceutical industries. At present, food-grade protease is the safest method to prepare active peptides. Therefore, in the present study, the enzymatic method was employed to hydrolyze β-LG to prepare the T cell epitope peptide with oral tolerance.

Gouw et al. [17] found that a T cell epitope at AA13~48 was identified in the infant formula supplemented with hydrolyzed whey powder using a T-cell proliferation assay; moreover, peptides that function as T cell epitopes were explored to support the development of oral tolerance. In this study, proteases which can induce T cell epitope release from β-LG were originally screened, and of these, trypsin is the optimal enzyme because of its hydrolysate with abundant T cell epitopes. Moreover, the hydrolysate of trypsin was added to whey protein to successfully prepare a product with oral immune tolerance. Koko Mizumachi et al. [23] studied the oral tolerance of AA42~56, AA62~76 and AA139~154 polypeptides and showed that these three peptides successfully inhibited T cell proliferation. Furthermore, AA139~154 successfully inhibited the production of β-LG antibodies, i.e., decreased the specific IgE and IgG_1_ antibody levels, and effectively induced the inhibition of Th2 subsets. Additionally, the amino acid sequence 13–48 of mature β-LG was noted to significantly reduce the acute allergic skin response in a murine model for cow milk allergies [17]. In addition, two T cell dominant peptides in AA 83–104 and AA155–169 of β-LG reduced the allergen-specific systemic markers in BLG-sensitized mice but failed to induce CD25+ Foxp3+ regulatory T cells in therapeutic studies [24]. Sophie et al. found that the β-LG tolerogenic fragment was enriched in the internal 3 tryptic peptides (84~100, 125~138, (61,62–69): S-S:(149–162)) of the β-LG molecule by animal experiments and by ELISA inhibition experiments [25]. However, not all T cell epitope peptides could induce immune tolerance. For instance, Meulenbroek et al. [26] concluded that T cell epitope peptides AA92~100 and AA91~108 had no tolerance. In the present study, a series of T cell epitope peptides from different β-LG hydrolysates were analyzed and preliminarily identified through the peripheral blood T lymphocytes of BALB/C mice sensitized with β-lactoglobulin. For the first time, some T cell epitope peptides of β-lactoglobulin were identified, such as AA100~107, AA92~107, AA31~56, AA108~116, AA108~117, AA57~76, AA146~162, AA138~162, AA104~111, AA112~119, AA111~119 and AA110~119. Further research is needed, such as verification of peripheral blood T cells of allergic patients or revealing the immune tolerance mechanism of T cell epitope peptides through animal experiments. Peptides with immune tolerance have broad application prospects. A drug had been developed that is useful for preventing or treating acute and chronic inflammation of the upper and lower respiratory tract, allergies and mixed etiologies, and it is comprised of alloferon peptides or alloferon peptide complexes [27].

The mechanism of a food allergy involves the stimulation of CD4+ T cells by food allergens, most often followed by differentiation into Th2 cells, making the Th2 cells dominate. Th2 cells produce cytokines IL-4, IL-5 and IL-13, which induce B cells to generate antigen-specific IgE antibodies. These antibodies bind to mast cells or basophils to stimulate their degranulation and the release of histamine and other active media, leading to allergic symptoms [28]. Additionally, in food allergies, chitinase-3-like protein 1 plays a major role in Th2 cell inflammation through phosphorylation of protein kinase B [29]. In addition, cytokines secreted by the Th1 cells include IFN-γ and IL-17, which can inhibit the differentiation of Th2 cells, thus alleviating allergic symptoms [30]. Inducing the production of allergen-specific IgG_1_ is one of the characteristics of an allergic Th2 reaction, which contributes to the production of Th2 cytokine [31,32]. Several studies have confirmed that higher expression levels of IL-4 and IL-5 will induce allergic diseases in infants [33]. The activated Th1 cells and CD8+ T cells can produce IFN-γ, which can promote the differentiation of undifferentiated T cells into Th1 cells and inhibit Th2 cell proliferation [34]. Studies have proved that Tregs are related to tolerance induction, and oral tolerance induction is probably due to the deletion of antigen-specific Th2 cells and the functional inhibition [35,36] of generated CD4+ CD25+ Foxp3+ Tregs and cytokines TGF-β and IL-10. In this study, the β-LG hydrolysates of trypsin, papain and protamex induced a significant increase in CD4+ CD25+ Foxp3+ Tregs in mice, which partially verified this conclusion. We also detected the levels of Th1 cell-related factors (IL-17 and IFN-γ), Th2 cell-related factors (IL-4, IL-5 and IL-13) and other factors associated with allergic symptoms, such as histamine, chitinase-3-like protein 1 and specific antibodies IgE, IgG_1_ and IgG_2a_, to verify whether the mice presented allergies or tolerance. The results showed that the hydrolysates of the TH, PM and PH groups induced oral tolerance in mice. Notably, this is the first study to report that the hydrolysates of protamex and papain have oral tolerance. As for the neutrase hydrolysate, no oral tolerance was observed in mice. Further research on the intestinal sensitization mechanism is highly recommended for the future. Gouw et al. [17] found that the whey protein hydrolysate containing T-cell epitopes could induce oral tolerance in mice. In contrast, our study results demonstrate that the hydrolysate containing T cell epitopes does not necessarily induce oral tolerance. Animal experiments are needed for further verification. In addition, we found that the variety of T cell epitope peptides in the hydrolysate might play an important role in whether the hydrolysate had immune tolerance properties. The mechanism behind oral immune tolerance induction by T cell epitopes has yet to be further revealed. Allergen-specific immunotherapy is emerging as a viable option for human desensitization, and oral immunotherapy using immunodominant peptides show therapeutic potential. However, long-term efficacy remains to be studied because the need for immunodominant peptides could be different among patients due to genetic HLA diversity [37].

## 5. Conclusions

A strategy for the prediction, identification and application of β-lactoglobulin hydrolysates with oral immune tolerance was established using bioinformatics, hydrolysis, mass spectrometry, T cell proliferation tests and animal experiments. The trypsin hydrolysate of β-lactoglobulin with abundant T cell epitopes exhibits significant oral tolerance, which could be used for the prevention of allergies in the future. In addition, it was initially found that β-lactoglobulin hydrolysates prepared by papain and protamex proteases contained less T cell epitope peptides, but had good oral tolerance activities. Although the neutral protease hydrolysate contained T cell epitopes, it did not induce immune tolerance in mice. 

## Figures and Tables

**Figure 1 foods-12-00307-f001:**
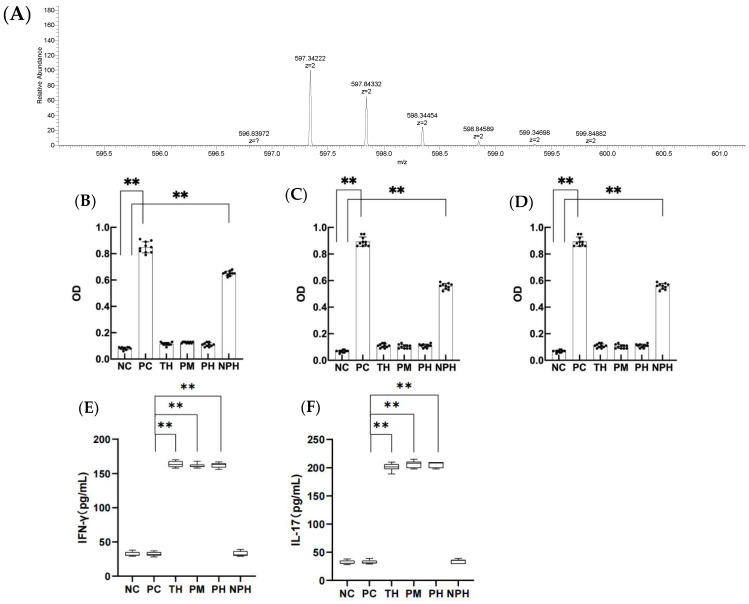
The primary mass spectra of DEALEKFDKALKALPMHIRLS peptide, (**A**) levels of specific IgE, IgG_1_ and IgG_2a_ in the serum of mice in each group (**B**–**D**) and contents of IFN-γ and IL-17 in the mice of each group (**E**,**F**). (**B**) corresponds to the level of specific IgE; (**C**) corresponds to the level of specific IgG_1_; (**D**) corresponds to the level of specific IgG_2a_; (**E**) corresponds to the level of IFN-γ; (**F**) corresponds to the level of IL-17. Data are shown as the mean ± SEM (n = 10). Each dot represents an individual mouse. ** means differs significantly (*p* < 0.05). NC corresponds to normal saline negative control group; PC corresponds to β-lactoglobulin positive control group; TH corresponds to trypsin hydrolysate group; PM corresponds to protamex hydrolysate group; PH corresponds to papain hydrolysate group; NPH corresponds to neutrase protease hydrolysate group.

**Figure 2 foods-12-00307-f002:**
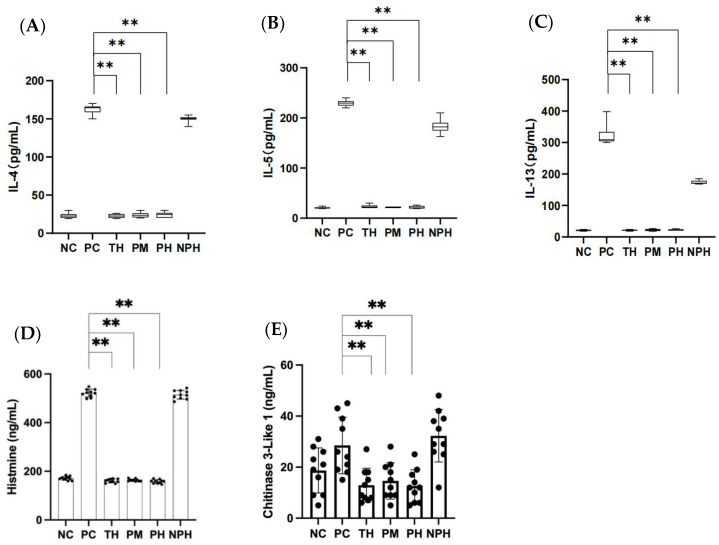
Contents of IL-4, IL-5, IL-13 (**A**–**C**), histamine (**D**) and chitinase-3-like protein 1 (**E**) in the mice of each group. (**A**) Corresponds to the level of IL-4; (**B**) corresponds to the level of IL-5; (**C**) corresponds to the level of IL-13; (**D**) corresponds to the level of histamine; (**E**) corresponds to the level of chitinase-3-like protein 1. Data are shown as the mean ± SEM (n = 10). ** means differs significantly (*p* < 0.05). NC corresponds to normal saline negative control group; PC corresponds to β-lactoglobulin positive control group; TH corresponds to trypsin hydrolysate group; PM corresponds to protamex hydrolysate group; PH corresponds to papain hydrolysate group; NPH corresponds to neutrase protease hydrolysate group. The black dots represent data (n = 10).

**Figure 3 foods-12-00307-f003:**
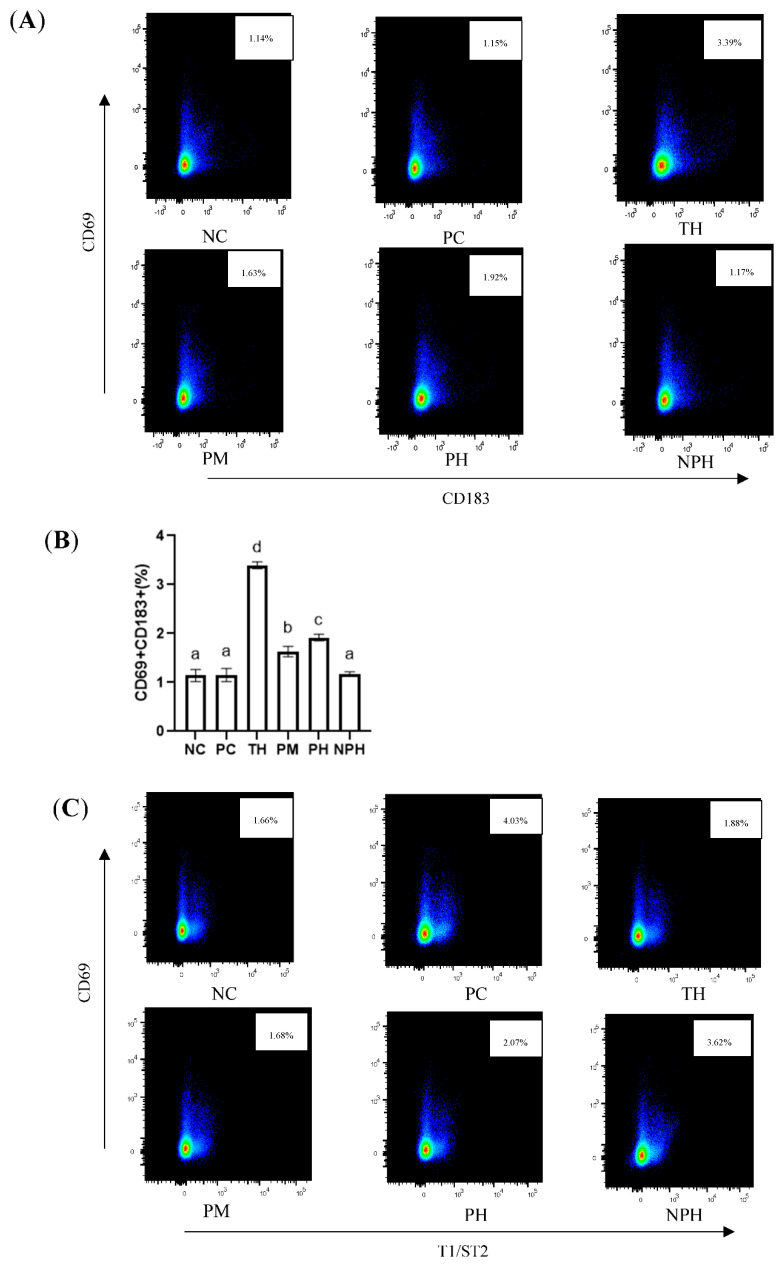
Differentiation levels of Th1 cells (**A**,**B**), Th2 cells (**C**,**D**) and Treg cells (**E**,**F**) in the spleen of Balb/c mice. NC corresponds to normal saline negative control group; PC corresponds to β-lactoglobulin positive control group; TH corresponds to trypsin hydrolysate group; PM corresponds to protamex hydrolysate group; PH corresponds to papain hydrolysate group; NPH corresponds to neutrase protease hydrolysate group. Data are shown as the mean ± SEM (n = 10) and different letters (a~d in (**B**); a~e in (**D**); a~b in (**F**)) represent significant differences (*p* < 0.05).

**Figure 4 foods-12-00307-f004:**
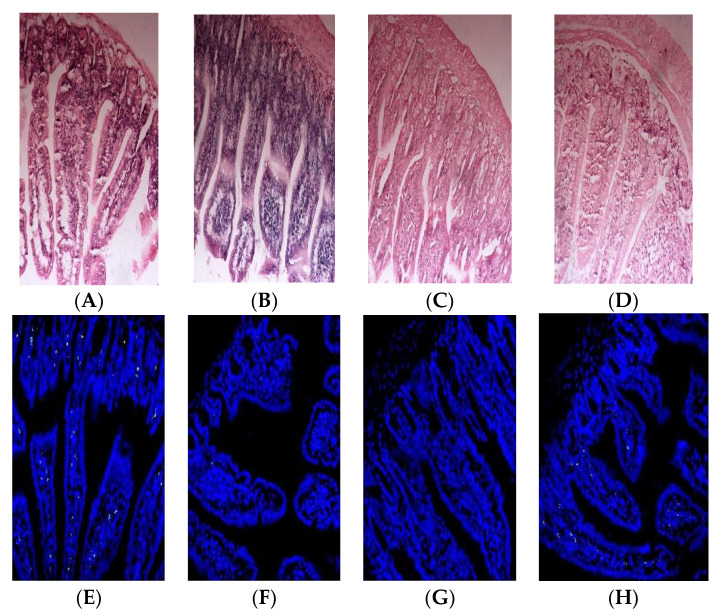
Histopathological observation of small intestines (**A**–**D**) and apoptosis results of intestinal villi epithelial cells (**E**–**H**) in Balb/c mice. (**A**,**E**) corresponds to β-lactoglobulin positive control group (PC); (**B**,**F**) corresponds to normal saline negative control group (NC); (**C**,**G**) corresponds to the TH group with early gastric administration of β-lactoglobulin trypsin hydrolysate (TH) added to whey protein concentrate; (**D**,**H**) corresponds to the mice group with early gastric administration of whey protein concentrate (WPC).

**Table 1 foods-12-00307-t001:** Amino acid sequence of β-lactoglobulin hydrolysate peptides and identification of T cell epitopes.

Name of Protease	Determination of Amino Acid Sequences of Peptides by Mass Spectrometry	Partial or Total Repeats with Predicting T Cell Epitopes	Potential T Cell Epitopes and the Sequence Location of β-Lactoglobulin	Stimulation Index by T Cell Proliferation Assay (SI)
Trypsin	ALKALPMH ALPMH IDALNENK IIAEKTKIPAVFK TKIPAVFK TKIPAVFKIDALNENK TPEVDDEALEK TPEVDDEALEKFDK TPEVDDEALEKFDKALK VAGTWYSLAMAASDISLLDAQSAPLR VLVLDTDYK VLVLDTDYKK VYVEELKPTPEGDLEILLQK	LKALPMH	ALKALPMH (AA155~162)	2.63 ± 0.07
		ALPMH	ALPMH (AA158~162)	2.12 ± 0.02
		IDALNENK	IDALNENK (AA100~107)	2.27 ± 0.03
		KTKIPAVFK	IIAEKTKIPAVFK (AA87~99)	2.53 ± 0.01
		TKIPAVFK	TKIPAVFK (AA92~99)	2.41 ± 0.02
		IPAVFKIDALNENK	TKIPAVFKIDALNENK (AA92~107)	3.12 ± 0.03
		VAGTWYSLAMAAS-DIS	VAGTWYSLAMAASDISLLDAQSAPLR (AA31~56)	2.64 ± 0.02
		DTDYK	VLVLDTDYK (AA108~116)	2.12 ± 0.01
		DTDYKK	VLVLDTDYKK (AA108~117)	2.04 ± 0.02
		VYVEELKPTPE	VYVEELKPTPEGDLEILLQK (AA57~76)	2.68 ± 0.02
Protamex	DEALEKFDKALKALPMH DISLLDAQSAPLR DISSLLDAQSAPLRVY KVAGTWYS LDIQKVAGTWYS LLDAQSAPLR LIVTQTMKGLDIQKVAGTWYS LVRTPEVDDEALEKFDKALKALPMH	KALKALPMH	DEALEKFDKALKALPMH (AA146~162)	2.96 ± 0.03
		KVAGTWYS	LDIQKVAGTWYS (AA26~37)	2.31 ± 0.02
		LIVTQTMKGLDIQ	LIVTQTMKGLDIQKVAGTWYS (AA17~37)	2.59 ± 0.03
		PMH	LVRTPEVDDEALEKFDKALKALPMH (AA138~162)	2.11 ± 0.02
Papain	LDAQSAPLR NENKVLVL	NENKV	NENKVLVL (AA104~111)	2.05 ± 0.02
Neutrase	DTDYKKYL KPTPEGDLEILL LDAQSAPLRV LDTDYKKYL VLDTDYKKYL VRTPEVDDEAL	DTDYKKYL	DTDYKKYL (AA112~119)	2.12 ± 0.01
		KPTPEGD	KPTPEGDLEILL (AA63~74)	2.34 ± 0.02
		LDTDYKKYL	LDTDYKKYL (AA111~119)	2.65 ± 0.02
		VLDTDYKKYL	VLDTDYKKYL (AA110~119)	2.56 ± 0.01

**Table 2 foods-12-00307-t002:** Clinical symptoms and scores for each group of mice.

Score	Symptoms	NC Group (n = 10)	PC Group (n = 10)	TH Group (n = 10)	PM Group (n = 10)	PH Group (n = 10)	NPH Group (n = 10)
0	No symptoms	10	0	9	8	8	0
1	Scratching nose and mouth	0	0	1	2	2	0
2	Edema around eyes and mouth; decreased activity; increased respiratory rate	0	0	0	0	0	2
3	Tachypnea; rashes surround mouth and tail; increased respiratory rate	0	6	0	0	0	6
4	No activity after stimulation with trembling and muscle contraction	0	3	0	0	0	2
5	Shock and death	0	1	0	0	0	0
Scoring		0	35.00 ± 0.56 a	1.00 ± 0.05 b	2.00 ± 0.08 b	2.00 ± 0.05 b	30.00 ± 0.12 a

Note: different lowercase letters represent significant differences between mice groups (*p* < 0.01).

## Data Availability

Data is contained within the article or supplementary material.

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
