# Peer review of "Preparation, Identification and Application of β-Lactoglobulin Hydrolysates with Oral Immune Tolerance"

_foods, 2023, doi:10.3390/foods12020307_

Round 1

Reviewer 1 Report

- The identification of "tolerogenic" peptides are of interest for allergic people.

Major comments

Introduction

- There are some misleading sentences, regarding T cell binding, oral tolerance, allergenicity. For example, line 31: what is a "pathogenic" peptide ? 

Materials and Methods

What was the dose of peptides used for the mice model ?Since the different enzymes did not lead to the same amount of peptides, nor the same length and so MW, how was it ensured that mice from different groups received the same amount of peptides ?

- For the mice model, the peptides are described to be goving with Whey 3%. What is the impact of whey alone on the outcomes ? A group with whey 3% without peptide is missing in the design of the experiment

- The proliferation assay does not give any info on the tolerogenic capacity of the peptide but only show they are recognized by T cell and induce proliferation. Line 82-83 and line 88

- Why was mice T cells used for the proliferation assay ? It would have been much more relevant to use T cells from milk allergic donors as in Gouw et al. 2018.

Results

- What is the exact message conveyed by Fig 1A ?

- All results from the animal model should be regrouped for the sake of clarity and easier reading

- Fig.1 : ratio between Th1 and Th2 would be important to show as well (IgG1/IgG2a)

- 3.5: the determination of the amount of specific IgE, IgG1 and IgG2 is not reflecting an allergic reaction as mentioned line 228 but sensitization to BLG

- Fig 4. the quality of the pictures should be improved

Discussion

- New opportunities for inducing immune unresponsiveness or tolerance are under development and should be mentioned in the paper

Minor comments

Introduction

- Intro line 37 "within 1 hour (rapid onset) or 1 hours (delayed onset), should be corrected. Delayed onset is 24-72h after ingestion

Materials and Methods

- Line 82: what does "some or all repeated polypeptides" were synthetized ?

- What is used in the different experiments and the name of the groups needs to be clarified. For in vitro: is it synthetic peptides ? For in vivo: is it synthetic peptides added to Whey 3% or hydrolysates  (as mentioned in legend of Fig 4). There is a lot on unclarity while this is highly important to interpret the data

Results

- Line 193: please spell our SI

- Line 253: mast cells are in tissue and basophils are in blood. The sentence should be corrected

- Legend of Fig.4 Why "trypsin hydrolysate group" is now mentioned as WPC ? Was it not TH earlier in the paper ? 

Discussion

- Line 322: the paper cited (ref 19) is not mentioning that "the polypeptide produced by protein hydrolysis is superior to protein itself". I mentioned that peptides with bioactivity can be generated from protein hydrolysis. This is a general statement that is not supported by evidence as as far as I am aware of.

- The authors should clearly state what is new in their paper as compared to other papers cited (Adel-Patient, Grouw, ...). A clear comparison with the peptides already described as promoting oral tolerance to BLG should be provided.

- New opportunities for inducing immune unresponsiveness or tolerance are under development and should be mentioned in the paper

- Attempts to use peptides in respiratory allergy have also already been tried, and not so successful so far and should be mentioned

- Hypothesis explaining the differences observed between the different peptides efficacy should be provided. Three out of the 4 Neutrase peptides contains one peptide from Trypsin (DTDYKK). Neutrase peptides are not efficient in the mouse model while Trypsin are. What to explain it ? Is it a matter of dose ? Of variety of peptides needed ?

Author Response

Introduction

- There are some misleading sentences, regarding T cell binding, oral tolerance, allergenicity. For example, line 31: what is a "pathogenic" peptide?

Answer: revised. Shown in lines 57-58, lines 64-65, lines 69-70.

Materials and Methods

- What was the dose of peptides used for the mice model? Since the different enzymes did not lead to the same amount of peptides, nor the same length and so MW, how was it ensured that mice from different groups received the same amount of peptides ?

Answer: Thank you for asking such a profound question. We had optimized the hydrolysis conditions. The peptide contents of Protease M, Alcalase, neutrase, protamex, papain and trypsin hydrolysates were respectively 485.12 ± 0.45 mg/g, 494.89 ± 1.24 mg/g, 515.78 ± 0.45 mg/g, 516.02 ± 0.78 mg/g, 516.23 ± 1.56 mg/g and 517.04 ± 0.85 mg/g using the biuret method, and the contents of peptides in different hydrolysates were not significantly different (p>0.05), so it is true that different groups received the same amount of peptides. Determination and result of peptide content of each hydrolysate were added in the article. Shown in lines 123-131, lines 296-300. We are so sorry for our carelessness, now these data are supplemented.

- For the mice model, the peptides are described to be goving with Whey 3%. What is the impact of whey alone on the outcomes? A group with whey 3% without peptide is missing in the design of the experiment

Answer: Now, a group with whey protein powder (WPC) without peptides is added in the article. In this study, the design of the experiment included a group with whey protein powder without peptides. Because the intestinal section results have not yet come out when we submit this article, so these data were not written when we submitted the article, and now it is added. Thank you for your understanding. Shown in lines 256-261, lines 400-412, lines 693-696, line 703.

- The proliferation assay does not give any info on the tolerogenic capacity of the peptide but only show they are recognized by T cell and induce proliferation. Line 82-83 and line 88

Answer: We carried out the proliferation assay according to the reference ( J.W. Gouw, J. Jo, L.A.P.M. Meulenbroek, T.S. Heijjer, E. Kremer, E. Sandalova, A.C. Knulst, P. v. Jeurink, J. Garssen, A. Rijnierse, L.M.J. Knippels, Identification of peptides with tolerogenic potential in a hydrolysed whey-based infant formula, Clinical and Experimental Allergy. 48 (2018) 1345–1353. https://doi.org/10.1111/cea.13223.)

Sentences that are incorrectly expressed had been modified. Shown in lines 151-154, lines 159-160.

- Why was mice T cells used for the proliferation assay? It would have been much more relevant to use T cells from milk allergic donors as in Gouw et al. 2018.

Answer: It is too difficult to find a milk allergic donor. It is correct that the mice T cells from the allergic mice were used for the proliferation assay. Thank you!

Results

- What is the exact message conveyed by Fig 1A ?

Answer: The peptides in the hydrolysate were all analyzed by mass spectrometry, and only the primary mass spectrum of VLVLDTDYKK polypeptide was placed in Figure 1 (A) due to limited layout. The signal peptides of the polypeptides put at the time of submission have not been removed, and they have now been modified. Thank you!

- All results from the animal model should be regrouped for the sake of clarity and easier reading

Answer: the methods and results of animal model had been revised. We are very sorry for our carelessness, and sentences or units that express inaccuracies have been modified. Shown in lines 256-261, lines 400-412.

- Fig.1 : ratio between Th1 and Th2 would be important to show as well (IgG1/IgG2a)

Answer: In the present study, the results of differentiation of TH1, TH2 and Treg cell subsets had shown that β-Lactoglobulin hydrolysate induces oral tolerance by regulating Thl/Th2 balance in splenic lymphocytes of the allergic mice.

- 3.5: the determination of the amount of specific IgE, IgG1 and IgG2 is not reflecting an allergic reaction as mentioned line 228 but sensitization to BLG

Answer: Thank you for the meticulous review, I have corrected the sentence that is inaccurate. Shown in line 342.

- Fig 4. the quality of the pictures should be improved

Answer: revised, and shown in Fig 4.

Discussion

- New opportunities for inducing immune unresponsiveness or tolerance are under development and should be mentioned in the paper

Answer: The outlook and future development about oral tolerance have been added to the discussion.

“Allergen-specific immunotherapy is emerging as a viable option for human desensitization, and oral immunotherapy using immunodominant peptides show therapeutic potential, but long-term efficacy remains to be studied because the need for immunodominant peptides could be different among patients due to genetic HLA diversity.” Shown in lines 493-497.

Minor comments

Introduction

- Intro line 37 "within 1 hour (rapid onset) or 1 hours (delayed onset), should be corrected. Delayed onset is 24-72h after ingestion

Answer: revised, and shown in lines 77-78.

Materials and Methods

- Line 82: what does "some or all repeated polypeptides" were synthetized ?

Answer: revised. Some polypeptides were synthetized. Shown in lines 151-152.

- What is used in the different experiments and the name of the groups needs to be clarified. For in vitro: is it synthetic peptides ? For in vivo: is it synthetic peptides added to Whey 3% or hydrolysates  (as mentioned in legend of Fig 4). There is a lot on unclarity while this is highly important to interpret the data

Answer: Firstly, we found β-LG hydrolysates had some peptides by MS, and these peptides may be potential T cell epitopes, then we synthesize these peptides according to the reference (K. Adel-Patient, S. Nutten, H. Bernard, R. Fritsché, S. Ah-Leung, N. Meziti, G. Prioult, A. Mercenier, J.M. Wal, Immunomodulatory potential of partially hydrolyzed β-lactoglobulin and large synthetic peptides, J Agric Food Chem. 60 (2012) 10858–10866. https://doi.org/10.1021/jf3031293), some synthetic peptides were identified as T cell epitopes. Taking into account practical application, the trypsin hydrolysates rich in T cell epitopes were added to whey protein powder, in addition, oral tolerance of whey powder containing the trypsin hydrolysates were identified by animal experiment. The legend of Fig 4 showed “added with β-lactoglobulin trypsin hydrolysate”. Every step of the experiment is scientifically reasonable, with literature and theoretical basis.

Results

- Line 193: please spell our SI

Answer: revised, stimulation index (SI), shown in line 312.

- Line 253: mast cells are in tissue and basophils are in blood. The sentence should be corrected

Answer: revised, and shown in line 358.

- Legend of Fig.4 Why "trypsin hydrolysate group" is now mentioned as WPC ? Was it not TH earlier in the paper ?

Answer: revised. Fig 4 “trypsin hydrolysate group” is mentioned as TH.

Discussion

- Line 322: the paper cited (ref 19) is not mentioning that "the polypeptide produced by protein hydrolysis is superior to protein itself". I mentioned that peptides with bioactivity can be generated from protein hydrolysis. This is a general statement that is not supported by evidence as as far as I am aware of.

Answer: The incorrect sentence had been deleted. Thank you!

- The authors should clearly state what is new in their paper as compared to other papers cited (Adel-Patient, Grouw, ...). A clear comparison with the peptides already described as promoting oral tolerance to BLG should be provided.

Answer: For the first time, some T cell epitope peptides of β -lactoglobulin were identified, such as AA100~107, AA92~107, AA31~56, AA108~116, AA108~117, AA57~76, AA146~162, AA138~162, AA104~111, AA112~119, AA111~119, AA110~119. Further research must be needed, such as verification with peripheral blood T cells of allergic patients or revealing the immune tolerance mechanism of T cell epitope peptides through animal experiments. Shown in lines 448-454.

- New opportunities for inducing immune unresponsiveness or tolerance are under development and should be mentioned in the paper

Answer: The outlook and future development about oral tolerance have been added to the discussion.

“Allergen-specific immunotherapy is emerging as a viable option for human desensitization, and oral immunotherapy using immunodominant peptides show therapeutic potential, but long-term efficacy remains to be studied because the need for immunodominant peptides could be different among patients due to genetic HLA diversity.” Shown in lines 493-497.

- Attempts to use peptides in respiratory allergy have also already been tried, and not so successful so far and should be mentioned

Answer: A patent suggests that some peptides are already being used in respiratory allergy treatment. The information about peptides used to treat respiratory allergy had been added in discussion. Shown in lines 454-457.

- Hypothesis explaining the differences observed between the different peptides efficacy should be provided. Three out of the 4 Neutrase peptides contains one peptide from Trypsin (DTDYKK). Neutrase peptides are not efficient in the mouse model while Trypsin are. What to explain it ? Is it a matter of dose ? Of variety of peptides needed ?

Answer: The hypothesis had been added in the discussion. “In addition, we found that the variety of T cell epitope peptide in the hydrolysate might play an important role to whether the hydrolysate had immune tolerance properties. The mechanism behind oral immune tolerance induction by T cell epitopes needs to be further revealed.” Shown in lines 490-492.

Reviewer 2 Report

The authors investigated the effects of β-lactoglobulin (β-LG) hydrolyzed from different enzymes on the induction of oral immune tolerance, using as model Balb/c mice. Different cell assays, like cytokines release or IgE and IgG determination, were performed. Moreover, the T cell epitopes of β-LG were predicted by bioinformatics tool, and subsequently, the polypeptide amino acid sequences of β-LG hydrolysates were identified by mass spectrometry and matched with the bioinformatic analysis.

In general, the study lacks novelty and requests an English revision.

The abstract poorly describes the study. 

Table 1, as well as paragraph 2.1 in the Material and Methods section,  need to be revised. Indeed, seems that the authors used in the study, only the peptides identified by LC-MS analysis and validated by bioinformatic tool, and not the entire form of β-LG hydrolyzed.

Moreover, the results obtained are not supported by an adequate discussion. 

Author Response

Q: In general, the study lacks novelty and requests an English revision.

Answer: The innovation of this study is reflected in the following two aspects: First, for the first time, some T cell epitope peptides of β -lactoglobulin were identified, such as AA100~107, AA92~107, AA31~56, AA108~116, AA108~117, AA57~76, AA146~162, AA138~162, AA104~111, AA112~119, AA111~119, AA110~119. Second, a strategy for the prediction, preparation, identification and application of β-lactoglobulin hydrolysate with oral immune tolerance was established.

An English revision had been carried out for this article. Shown in lines 57-58, 64-65, 69-70, 77-78, 84, 88-94, etc.

Q: The abstract poorly describes the study.

Answer: revised. Shown in lines 31-32, lines 35-36.

Q: Table 1, as well as paragraph 2.1 in the Material and Methods section, need to be revised. Indeed, seems that the authors used in the study, only the peptides identified by LC-MS analysis and validated by bioinformatic tool, and not the entire form of β-LG hydrolyzed.

Answer: Table 1 had been revised. In present study, the amino acid sequence of peptides in β-LG hydrolysates were all identified by LC-MS. Subsequently, the sequences of hydrolysate polypeptide determined by mass spectrometry were compared with the predicted T cell epitopes. Some polypeptides were synthesized by solid phase synthesis method[14]. Then, T cell proliferation method was used to identify the T cell epitope of the synthetic polypeptide based on the modified methods of HOU et al.[15]

Because there are many peptides in each hydrolysate of β-LG, if all peptides are synthesized by solid phase synthesis method and then subjected to T cell proliferation experiments, it requires a lot of work and a lot of experimental expenses, and more importantly, the method of bioinformatics to predict T cell epitopes has been reported in many references, and this method shows high accuracy.

Q:Moreover, the results obtained are not supported by an adequate discussion.

Answer: The discussion section has been revised and some sentences and references have been added. Shown in lines 436-443, 448-451, 454-457,490-497.

Reviewer 3 Report

I have examined the paper. The topic is interesting but my concern is that it is very hard to follow. Information is very scattered and descriptions often seem to be incomplete. Most seems to be in the form of very short sections which make one or two statement, but the authors fail to build a real storyline that takes the reader along the journey through their paper. As a result, the reader has to ‘work very hard’ to actually get into the paper and comprehend what is reported. This really needs to be improved, otherwise the work will be of limited value to the community.

Overall, I also find characterization of the hydrolysates quite limited. %DH plus some mention of a few peptides does not really support the study. So, I would suggest this is improved. When using b-Lg from Sigma, I also feel obliged to ask the authors whether the know the purity of the material, as it is never 100%.

Table 1 seems to be impossible. I see e.g., positions 164, 166 etc for b-LG mentioned in the sequence allocation but b-Lg does not contain that many amino acids. Did you forget to subtract the signal peptide? This also goes for all specific mentions for amino aciss in text.

Table 2 and description: very observational. Need to see some statistical relevance here.

Author Response

Q:I have examined the paper. The topic is interesting but my concern is that it is very hard to follow. Information is very scattered and descriptions often seem to be incomplete. Most seems to be in the form of very short sections which make one or two statement, but the authors fail to build a real storyline that takes the reader along the journey through their paper. As a result, the reader has to ‘work very hard’ to actually get into the paper and comprehend what is reported. This really needs to be improved, otherwise the work will be of limited value to the community.

Answer: Thank you for your valuable comments. We have completely revised the summary, materials and methods, results, and discussion sections of the paper. Also, an English revision had been carried out. The revised section was shown in red font in the article.

Q:Overall, I also find characterization of the hydrolysates quite limited. %DH plus some mention of a few peptides does not really support the study. So, I would suggest this is improved. When using b-Lg from Sigma, I also feel obliged to ask the authors whether the know the purity of the material, as it is never 100%.

Answer: Thank you for your valuable comments. In present study, the amino acid sequence of peptides in β-LG hydrolysates were all identified by LC-MS. Subsequently, the sequences of hydrolysate polypeptide determined by mass spectrometry were compared with the predicted T cell epitopes. Some polypeptides were synthesized by solid phase synthesis method[14]. Then, T cell proliferation method was used to identify the T cell epitope of the synthetic polypeptide based on the modified methods of HOU et al. [15]. Moreover, the peptide content of each hydrolysate was supplemented.

The purity of β-Lg from Sigma ≥90 %. The purity of β-Lg is identified by electrophoresis, which is a single band suitable for hydrolysis experiments.

Q:Table 1 seems to be impossible. I see e.g., positions 164, 166 etc for b-LG mentioned in the sequence allocation but b-Lg does not contain that many amino acids. Did you forget to subtract the signal peptide? This also goes for all specific mentions for amino acids in text.

Answer: Thank you for your valuable comments. We apologize for our carelessness. Remove the signal peptide, and the last amino acid number of β-lactoglobulin is 162. The trypsin and protamex hydrolysates were analyzed by LC-MC again. And the peptides at AA 155-162, AA158-162, AA 146-162 and AA 138-162 were synthesized by solid phase synthesis method and identified by T cell proliferation method again. The data revised was shown in Table 1 and the text.

Table 2 and description: very observational. Need to see some statistical relevance here.

Answer: revised. The data were statistically analyzed in table 2.

Round 2

Reviewer 3 Report

I have a few comments left:

 Overall, the manuscript could really benefit from language editing to improve readability

Line 13: a bit strange to start a sentence this early in the abstract with ‘In conclusion,’. This is normally only appropriate for (one of) the last sentences

Line 223: “The oral immune tolerance function of whey protein powder (WPC) containing β-LG trypsin hydrolysate (TH) was identified by animal experiments” is very unclear. WPC is normally used as an abbreviation for whey protein concentrate, but from what I guess what you mean is the tryptic hydrolysate of b-LG? Or not? This needs to be clarified as it is very unclear at the moment.

Author Response

Reviewer 1

I have a few comments left:

Overall, the manuscript could really benefit from language editing to improve readability

Answer: The manuscript has been checked and revised by a native English-speaking colleague.

Line 13: a bit strange to start a sentence this early in the abstract with ‘In conclusion,’. This is normally only appropriate for (one of) the last sentences

Answer: revised.

Line 223: “The oral immune tolerance function of whey protein powder (WPC) containing β-LG trypsin hydrolysate (TH) was identified by animal experiments” is very unclear. WPC is normally used as an abbreviation for whey protein concentrate, but from what I guess what you mean is the tryptic hydrolysate of b-LG? Or not? This needs to be clarified as it is very unclear at the moment.

Answer: WPC is revised as an abbreviation for whey protein concentrate in the present study. The oral immune tolerance function of the tryptic hydrolysate of β-LG (TH) added to whey protein concentrate (WPC) was identified by animal experiments.

If one of the referees has suggested that your manuscript should undergo extensive English revisions, please address this issue during revision. We propose that you use one of the editing services listed at https://www.mdpi.com/authors/english or have your manuscript checked by a native English-speaking colleague.

Answer: The manuscript has been checked and revised by a native English-speaking colleague.